# Dual symplectic classical circuits: An exactly solvable model of many-body chaos

Alexios Christopoulos[1*], Andrea De Luca[1], Dmitry L. Kovrizhin[1] and Tomaž Prosen[2]

**1** Laboratoire de Physique Théorique et Modélisation, CY Cergy Paris Université,
CNRS, F-95302 Cergy-Pontoise, France
**2** Faculty of Mathematics and Physics, University of Ljubljana,
Jadranska 19, SI-1000 Ljubljana, Slovenia

⋆ alexios.christopoulos@cyu.fr

## Abstract

We propose a general exact method of calculating dynamical correlation functions in dual symplectic brick-wall circuits in one dimension. These are deterministic classical many-body dynamical systems which can be interpreted in terms of symplectic dynamics in two orthogonal (time and space) directions. In close analogy with quantum dual-unitary circuits, we prove that two-point dynamical correlation functions are non-vanishing only along the edges of the light cones. The dynamical correlations are exactly computable in terms of a one-site Markov transfer operator, which is generally of infinite dimensionality. We test our theory in a specific family of dual-symplectic circuits, describing the dynamics of a classical Floquet spin chain. Remarkably, expressing these models in the form of a composition of rotations leads to a transfer operator with a block diagonal form in the basis of spherical harmonics. This allows us to obtain analytical predictions for simple local observables. We demonstrate the validity of our theory by comparison with Monte Carlo simulations, displaying excellent agreement with the latter for a choice of observables.



## 1  Introduction

Symplectic dynamics is a powerful framework for understanding the behaviour of classical systems in a wide range of physical phenomena, from celestial mechanics to fluid dynamics. At its core, symplectic dynamics is concerned with the study of the evolution of systems that conserve phase space volume under Hamiltonian motion. This property is intimately related to the presence of a geometric structure known as a symplectic form, which encodes the essential dynamical information of the system. An example of this type of dynamics that has attracted a lot of interest appears in the studies of classical spin chains. In particular, integrability has been studied for the classical Heisenberg spin chain (CHSC) [1, 2] in the $SU(2)$ symmetric case as well as in its generalizations [3, 4]. In addition, ergodicity has been studied for various types of 1D classical spin chain models [5–7], as well as the way it breaks [8] depending on the range of the interactions.

Recently, the framework of fluctuating hydrodynamics [9], originally introduced in classical anharmonic chains, has been fruitful in the study of correlations [10–13] in classical ferromagnetic spin chains, wherein a suitable intermediate temperature regime, the system was observed to show Kardar-Parisi-Zhang (KPZ) scaling. Quantum correspondence with spin chains [14] has demonstrated that there is a good agreement in the high-temperature limit even when the system is far away from the large spin limit.

Dual symplectic dynamics is a novel idea according to which symplecticity characterizes both time and space propagation. This has been observed in $SO(3)$-invariant dynamics of classical spins [15], where the correlation function exhibits KPZ universality [10, 16–18], with the spin transport being characterised by a dynamical exponent of 3/2. There has been a lot of recent work on the quantum analogue of dual symplecticity, namely dual unitarity in brick-wall quantum circuits, where both space and time propagators are unitary.

Interestingly, dual unitary quantum circuits can exhibit strongly chaotic quantum dynamics whose classical simulation is, in general, expected to be exponentially hard in system size [19]. Remarkably, dual unitarity offers the possibility to calculate exactly certain dynamical quantities, such as space-time correlation functions [20–22], spectral form-factor [23, 24], operator entanglement, and entanglement growth [25, 26].

In this paper, we propose a general exact method of calculating dynamical correlation functions in dual symplectic brick-wall 1D circuits. We show, that similarly to what happens for dual-unitary quantum circuits, correlation functions in space and time over the equilibrium uniform measure of single-site observables $\langle O(x,t)O(0,0)\rangle$ are such that: i) vanish everywhere except the light rays $x = \pm t$; ii) their behaviour on the light rays can be expressed

in terms of the matrix elements of a transfer operator. We demonstrate that our theory is in excellent agreement with the numerical calculation on an example of a specific family of dual-symplectic spin chains, where local gates are composed of Ising Swap gates and one-site rotations. For this model, we prove that, despite the infinite dimensionality of the local phase space, the transfer operator involved in the calculation of the correlation functions splits into finite-dimensional blocks, owing to the conservation of the total angular momentum. Using this decomposition we obtain exact analytical expressions for some observables and implement a simple and efficient numerical procedure for general ones.

The paper is organised as follows: in Section 2, we set up the general formalism for a symplectic characterised by a finite measure phase space; in Section 3 we discuss the dual symplectic case, and using a graphical representation present exact expressions for the correlations of arbitrary local observables. In Section 4, we discuss an example of the application of our theory for the Ising Swap model on a spin chain and show how it can be solved by block-diagonalisation of the transfer operator using conservation of the total angular momentum.

## 2 The model

We consider a classical dynamical system of $N$ variables $\{\vec{X}_i\}$, with the site index $i=0,\ldots,N-1$. For simplicity, we take $N$ to be even. We also assume that dynamical variables live on the finite measure space $M$, and the phase space of the whole system is obtained as the product of its $N$ copies, i.e. $M_N = M \times \ldots \times M$. The time is considered to be discrete $t \in \mathbb{Z}$, and the interactions are local. In particular, we express the dynamics in terms of a local symplectic map acting on two sites only – the so-called (classical) gate, $\Phi : M \times M \to M \times M$. The dynamics of the whole system is then obtained by acting with $\Phi$ on all pairs of neighbouring sites according to the brick-wall circuit protocol, where we impose periodic boundary conditions $\vec{X}_{i+N} \equiv \vec{X}_i$ (see Fig. 1).

Specifically, let's denote the local gate as $\Phi_{ij} : M_N \to M_N$, which acts as the map $\Phi$ on the variables $\vec{X}_i, \vec{X}_j$, and trivially with respect to all other variables. We can then introduce $\mathcal{T}_{even} = \Phi_{0,1}\Phi_{2,3}\ldots\Phi_{N-2,N-1}$ and similarly $\mathcal{T}_{odd} = \Sigma^{-1}\circ\mathcal{T}_{even}\circ\Sigma$, with the single-site translation $\Sigma : M_N \to M_N$, defined as $\Sigma(\vec{X}_0,\vec{X}_1,\ldots\vec{X}_{N-1}) = (\vec{X}_1,\ldots,\vec{X}_{N-1},\vec{X}_0)$. From these two layers, we construct the *Floquet Operator* $\mathcal{T}$ which generates one period of the dynamics,

$$\mathcal{T} = \mathcal{T}_{odd} \circ \mathcal{T}_{even}. \tag{1}$$

It is clear that $\Sigma^{-2}\mathcal{T}\Sigma^2 = \mathcal{T}$, namely there is a two-site translation invariance of the dynamics. In the following we represent a point of $M_N$ with bold capital letters e.g $\boldsymbol{X} \equiv (\vec{X}_0,\vec{X}_1,\ldots\vec{X}_{N-1})$ whereas, a point of the single site space $M$ is represented with a vector e.g $\vec{X}$.

It is useful to introduce a graphical notation. Specifically, we represent the local gate as a blue rectangle with two incoming and two outgoing legs

$$\Phi = \quad\blacksquare\quad \tag{2}$$

Each leg represents a copy of $M$, and an operator has as many legs as the number of sites it acts on. With this in mind, the single time-step operator $\mathcal{T}$ is graphically depicted in Fig. 1.

The map $\Phi$ belongs to a special group of transformations called the symplectic group. Symplecticity is a property appearing in Hamiltonian systems because they preserve the loop action [27]. In general, symplectic maps always involve $d$-pairs of conjugate variables, the configuration $q$ and the momentum $p$, which can be seen as the coordinates of a $2d$ dimensional manifold $\mathcal{M}$ (phase space) endowed with a symplectic form $\omega$ [27] on $\mathcal{M}$. Then, a

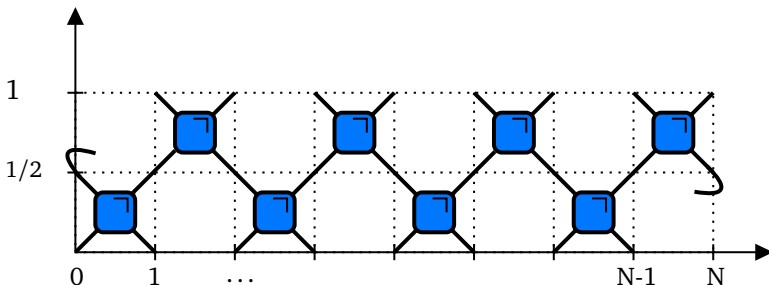

Figure 1: A graphical representation of the time evolution of a symplectic brick-wall circuit for a single tme-step.

symplectic map $g : \mathcal{M} \to \mathcal{M}$, must satisfy $Dg^T \omega Dg = \omega$ for the Jacobian matrix $Dg$ of the map $g$. Symplecticity implies a unit determinant of the Jacobian $\det(Dg) = 1$ and thus conservation of the phase space volume. However, symplecticity is more restrictive than just the conservation of the phase space volume as it also imposes restrictions on the spectrum. In particular, the spectrum $\sigma(Dg) = \{g_i\}_{i=1}^{2d}$ of the Jacobian includes only pairs of eigenvalues in the form $g_i, 1/g_i$ [27]. An important consequence of this property of $\sigma(Dg)$, is that the Lyapunov exponents $\lambda_i \quad i = 1, \ldots, 2d$ of the dynamics appear in pairs of $\pm \lambda_i$ [28].

## 3 Dynamical correlations

### 3.1 Symplectic gate

In this section, we will consider 2-point correlation functions, and show how symplecticity of the dynamics can be used to simplify the calculation of these functions. Before proceeding, we establish some definitions. First, we introduce the space of real functions over the phase space $M_N$

$$D(M_N) = \{\rho | \rho : M_N \to \mathbb{R}\}. \tag{3}$$

An important role is played by phase-space distributions in $D(M_N)$ satisfying

$$\rho(X) \in \mathbb{R}^+, \qquad \int dX \rho(X) = 1. \tag{4}$$

For technical reasons, it is however useful to consider the $L^2$ norm

$$\|\rho\| = \left[ \int dX |\rho(X)|^2 \right]^{1/2}, \tag{5}$$

and introduce a Hermitian product

$$\langle \rho_1 | \rho_2 \rangle = \int dX \, \rho_1^*(X) \rho_2(X), \qquad \rho_1, \rho_2 \in L^2(M_N), \tag{6}$$

with the bra-ket notation $\langle X | \rho \rangle = \rho(X)$. In general, any dynamical system with a map $h : M_N \to M_N$ on the phase space induces a dynamical transfer operator $\mathcal{P}_h : D(M_N) \to D(M_N)$. The map $\mathcal{P}_h$ is linear and is known as Frobenius-Perron operator [29] with a Dirac delta kernel

$$\mathcal{P}_h(X, Y) = \delta\big(X - h(Y)\big), \quad X, Y \in M_N, \tag{7}$$

and it performs the dynamics for given initial conditions (e.g. density).

In the case of the symplectic gate, which is invertible, the dynamical operator acts explicitly on the phase-space distribution $\rho$ as

$$(\mathcal{P}_\Phi \circ \rho)(X) = \int_{M_2} dY \, \delta\big(X - \Phi(Y)\big)\rho(Y) = \rho\big(\Phi^{-1}(X)\big), \quad X \in M_2 \,, \tag{8}$$

where we used the Jacobian of $\Phi$ being equal to one since the map is volume preserving. The additional structure of the Hilbert space can be exploited to represent $\mathcal{P}_\Phi$ as an infinite-dimensional unitary matrix. The unitarity $\langle \rho_1 | \mathcal{P}_\Phi^\dagger \mathcal{P}_\Phi | \rho_2 \rangle = \langle \rho_1 | \rho_2 \rangle$ follows from the volume preservation of the phase space.

An important consequence of symplecticity, is the invariance of the uniform (flat) measure on $L^2(M \times M)$ under the action of $\mathcal{P}_\Phi$. If we denote a single-site uniform measure as $u = 1/|M| \to |u\rangle$ with $|M|$ being the volume of the phase space for this site $M$, then we can construct the 2-site uniform measure as $|u\rangle \otimes |u\rangle$. Then, symplecticity implies, see Eq. (8), that any constant scalar is invariant under $\mathcal{P}_\Phi$ and so is the uniform density, which leads to the following equations

$$\mathcal{P}_\Phi |u\rangle \otimes |u\rangle = |u\rangle \otimes |u\rangle \,, \qquad \langle u| \otimes \langle u| \mathcal{P}_\Phi = \langle u| \otimes \langle u| \,, \tag{9}$$

where we use the fact that $\mathcal{P}_\Phi$ is a unitary operation in $L^2(M \times M)$ and thus the left and right eigenvectors are the same. It is convenient to work with the normalised state $|\circ\rangle = \|u\|_2^{-1}|u\rangle$ and choosing the graphical representation $|\circ\rangle = \overset{\mid}{\circ}$ so Eq. (9) is depicted as

$$\tag{10}$$

It is straightforward to check that this property implies that the stationary density of the Floquet transfer operator $\mathcal{T}$ is the uniform measure on $M_N$, which is denoted as $|u_N\rangle = |u\rangle \otimes \ldots \otimes |u\rangle$. We also introduce the $L^2$–normalised version $|\circ_N\rangle = \|u_N\|_2^{-1}|u_N\rangle$. Given any function on the phase space $a \in D(M_N)$, representing physical observable, we can express its average over the phase-space density $\rho$ as

$$\int dX \, a(X)\rho(X) = \langle 1_N | \hat{a} | \rho \rangle \,, \tag{11}$$

where the action of $\hat{a}$ is defined via $\langle X | \hat{a} | \rho \rangle = a(X)\rho(X)$, and we make use of the unit scalar $|1_N\rangle \to 1_N(X) = 1, \forall X \in M_N$. Note that we have $|1\rangle = \sqrt{|M|}|\circ\rangle$. In general, for an ergodic symplectic system $|\circ_N\rangle$ is the unique invariant measure and thus at long times, any initial state will always converge to that. In our setting, we consider correlations of observables at long times and thus we focus on the invariant uniform measure. The connected dynamical correlation functions for the one-site observables are defined as

$$C_{ab}(i,j;t) \equiv \langle 1_N | \hat{b}_j \mathcal{T}^t \hat{a}_i | u_N \rangle - \langle b_j \rangle \langle a_i \rangle = \langle \circ_N | \hat{b}_j \mathcal{T}^t \hat{a}_i | \circ_N \rangle - \langle \circ_N | \hat{b}_j | \circ_N \rangle \langle \circ_N | \hat{a}_i | \circ_N \rangle \,, \tag{12}$$

with $i,j = 0, \ldots, N-1$. In this expression, the local operators $\hat{a}_i$ act non-trivially only on the respective $i$-site, so that they can be expressed as $\hat{a}_i = \mathbb{1} \otimes \cdots \otimes \overset{\text{site } i}{\hat{a}} \otimes \cdots \otimes \mathbb{1}$. The second term is the product of the averages over the uniform measure which, for a local observable is defined as $\langle a_i \rangle = \langle 1_N | \hat{a}_i | u_N \rangle = \langle \circ | \hat{a} | \circ \rangle$.

In the following, we focus mainly on the nontrivial first term $\langle 1_N | \hat{b}_j \mathcal{T}^t \hat{a}_i | u_N \rangle$ of the correlations, which is shown in Fig. 2, where operations on a single site such as $\hat{a}|\circ\rangle$ or $\hat{b}|\circ\rangle$ are indicated with a bullet $\bullet$. Moreover, using the invariance of the circuit with respect to two-site

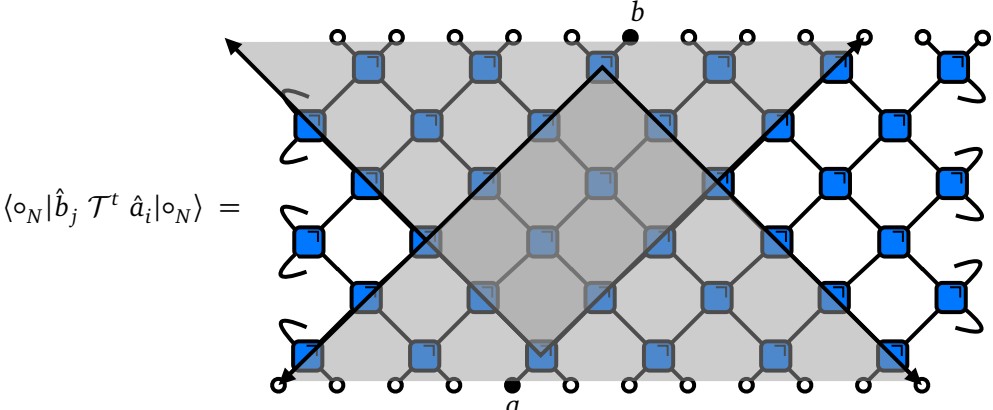

Figure 2: Graphical representations of the 2-point correlation function. The shaded grey areas and the black arrows indicate the causal cones attached to each local observable, with the "curly" edges indicating the periodic boundary conditions. The symplecticity of $\Phi$ reduces this circuit to the cross-section of the causal cones (double-shaded area of the grid).

shifts, one can map the correlations from point $i$ to 0 or 1 depending on the parity of $i$. This implies that the correlations split into two different types,

$$C_{ab}(i,j;t) = \begin{cases} C_{ab}^+(j-i;t), & i = \text{even}, \\ C_{ab}^-(j-i;t), & i = \text{odd}. \end{cases} \tag{13}$$

As we can see in Fig. 2, by applying Eq. (10), one can erase all gates outside of the light-cone, which spreads with velocity $v_c = 2$ from the position $i$ of the operator $\hat{a}$ at the bottom. One can use a similar argument starting from the top at the position $j$ of the operator $\hat{b}$. This suggests that the only remaining gates must lie in the intersection between the forward and backward light-cones (see Fig. 2). In particular, when $|i - j| > 2t$ then these light cones do not overlap, and the two observables are trivially uncorrelated. When $|i - j| \leq 2t$ the causal cones do overlap and can lead to non-trivially vanishing correlations. For times $t > N/4$ the light cones reach the boundary. This introduces finite-size effects, which makes the analytical calculations more complicated. Below we focus on times $t \leq N/4$ where the correlation functions are the same as in the thermodynamic limit $N \to \infty$. As explained above, we can make use of the symplecticity of the gate $\mathcal{P}_\Phi$ as expressed by Eq. (10) to cancel all gates outside the intersection of the two light-cones. This leads to the following representation

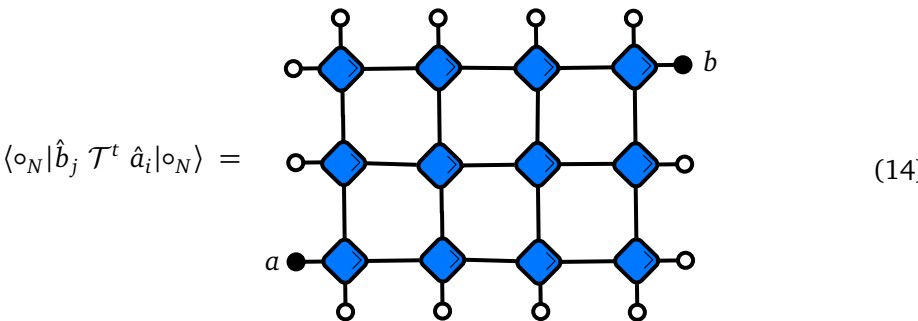

$$\langle \circ_N | \hat{b}_j \, \mathcal{T}^t \, \hat{a}_i | \circ_N \rangle = \tag{14}$$

where the diagram is rotated by 45° and we do not consider the case with the local observables on the same edge of the light-cone. The rectangle can be decomposed into rows or columns, which are represented as two different types of contracting transfer operators. This idea appears in the same manner in the folded picture of unitary circuits [20], and although

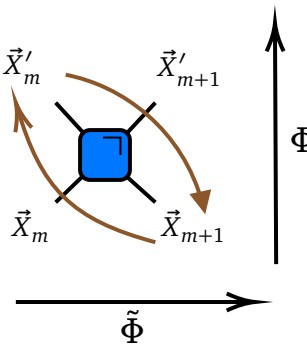

Figure 3: The local map $\Phi$ acting on two neighbouring spins and performing their temporal dynamics. By exchanging the diagonal legs we get the dual map $\tilde{\Phi}$ which performs spatial dynamics. The diagonal exchange of the legs exchanges the time and the space axes producing a map that propagates the temporal change in space.

it represents an important simplification, the calculation of 2-point correlation functions, still remains challenging particularly, when $|i-j|$ does not scale with $t$, because the size of the involved transfer operators grows with time. We will see in the following section that for dual symplectic gates additional simplifications are possible which allows one to calculate the correlation functions explicitly.

## 3.2 Dual-symplectic gates

So far, the discussion has been quite general. In order to, make explicit calculations here, we introduce an additional restriction to the dynamics. We demand that the local gate $\Phi$ is dual-symplectic. In other words, the evolution of the system remains symplectic when one exchanges the roles of space and time. Specifically, one can introduce the map performing the propagation in space, which is called the dual map $\tilde{\Phi}$. As in the case of dual-unitary circuits [25], it can be obtained by reshuffling diagonal legs as shown in Fig. 3, which leads to the exchange of time and space axes. In particular, one can see from Fig.3 that in the dual picture, the two adjacent times of one site define the same times of its neighbouring site on the right. One can also show, that for a dual-symplectic system knowing the time evolution of one site, one can uniquely determine the time evolution of the whole system. The dual picture allows for diagrams, like the one in Fig 2, to be interpreted in the space direction from left to right, with the exchange of $\Phi \rightarrow \tilde{\Phi}$ or from right to left, where the dual map is defined as in Fig. 3, but with the exchange of the legs of $\Phi$ with respect to the other diagonal. These diagrams are graphical representations of integrals over the phase space $M_N$ and the passing to the dual picture is a change of integration variables, which leads to a factor coming from the Jacobian of the transformation. In order, for both pictures to be equivalent under this change of variables, one has to ensure that the value of this Jacobian is equal to 1 for both left-to-right and right-to-left directions in space. Thus the local gate should satisfy the following conditions

$$\left|\det\left(\frac{\partial \Phi^1(\vec{X}_1,\vec{X}_2)}{\partial \vec{X}_2}\right)\right| = \left|\det\left(\frac{\partial \Phi^2(\vec{X}_1,\vec{X}_2)}{\partial \vec{X}_1}\right)\right| = 1, \quad \forall \vec{X}_1,\vec{X}_2 \in M \times M, \tag{15}$$

where $\Phi^{1,2}$ are the single-site outputs of the local gate defined as $\left(\Phi^1(\vec{X}_1,\vec{X}_2),\Phi^2(\vec{X}_1,\vec{X}_2)\right) = \Phi(\vec{X}_1,\vec{X}_2)$. We provide an explicit proof of (15) in the Appendix A. In addition, the dual map, by definition, is an involution and so the dual of the dual picture should be the original one with $\Phi$. In order to assure that the change from the original picture, in the time direction, to the dual one, and vice versa, is equivalent the Eq, (15) should respectively hold for the dual map. We prove in Appendix A that the condition (15) for $\Phi$ is sufficient for this to hold.

In general, an arbitrary symplectic map typically has a dual-space propagator which is not unique (non-deterministic) or not even defined for all points in $M \times M$. Here we focus on a local gate $\Phi$ with a uniquely defined and symplectic $\tilde{\Phi}$, that also satisfies (15). We stress that Eq. (15) is crucial and follows naturally in dual symplectic circuits which are obtained through a limiting procedure from dual-unitary quantum circuits with a finite discrete local Hilbert space. In fact, there has already been some work on dual symplectic circuits where Eq. (15) does not hold. In particular, in integrable circuits with non-abelian symmetries, it has been demonstrated [15] that 2-point dynamical correlations follow Kardar-Parisi-Zhang (KPZ) universality and are not restricted to the edges of the light-cone. This is in contrast with what we prove here for the dual-symplectic circuits where Eq. (15) holds.

With the additional property of dual symplecticity, the set of graphical contraction rules (10) is extended as

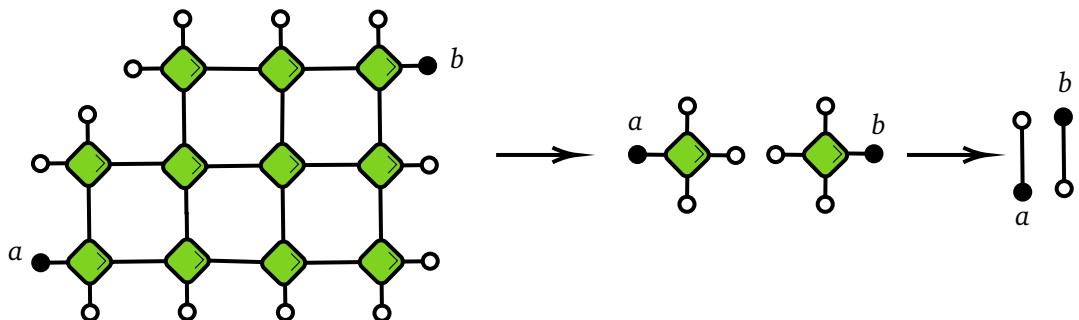

$$(16)$$

where the dual-symplectic gates are now being indicated with green colour. Dual symplecticity ensures the invariance of the uniform measure in the space direction as well. Its analogue in quantum systems is called dual unitarity and has been used to obtain exact results for a number of different systems [23,30,31]. There are similar expectations for dual-symplectic dynamics, and indeed in the following we show that one can use dual-symplecticity, to obtain exactly the dynamical correlation functions, and to show that they do not vanish only along the edges of the causal cones with (13) becoming

$$C_{ab}(i,j;t) = \begin{cases} \delta_{j-i,2t} \, C_{ab}^+(2t;t), & i = \text{even}, \\ \delta_{j-i,-2t} \, C_{ab}^-(-2t;t), & i = \text{odd}. \end{cases} \qquad (17)$$

We are going to prove this, using the diagrammatic representation established above. Specifically, one can simplify the correlations depicted in (14), by applying (16) at the edge of the rectangular area with neighbouring $|\circ\rangle$ states.

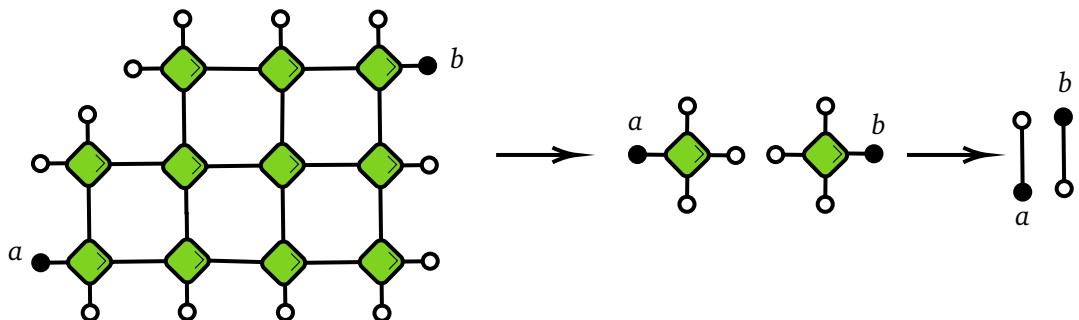

Repeating this process, we observe that the diagram trivialises to the second term of (12) and thus the connected correlations vanish. As long as this type of edge exists, the correlations will vanish except, when the surface area of the cross-section is zero and the parities of the sites of the local observables are the same. This would imply, that either of the sides of the cross-section has length zero, and the rectangle reduces to just a line segment of length $2t$,

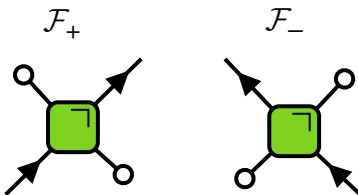

Figure 4: The graphical representation of two different types of transfer operators $\mathcal{F}_{\pm}$. On the left (right) is the transfer operator appearing on the right (left) moving light edge on (18).

with the local observables at the edges. From Fig. 2, one can see that depending on the parity of the site $i$ there are two different types of line segments

$$C_{a,b}^{+}(v_c t, t) = \quad a \qquad \qquad b \qquad i = \text{even},$$

(18)

$$C_{a,b}^{-}(-v_c t, t) = \quad b \qquad \qquad a \qquad i = \text{odd}.$$

When $i$ is even, the correlations survive along the right-moving light edge, and when $i$ is odd it is the same for the left-moving edge. In fact, one only has to study correlations of single chirality, since the correlations with opposite chirality, can be obtained via reflection of the circuit. As can be seen in Fig. 2, a reflection with respect to the axis passing between the points $(N/2-1, N/2)$ (which implies that every site $i = 0, \ldots, N-1$ is mapped to $N-1-i$), exchanges the two edges of the causal-cone. Furthermore, this reflection does not only change the parity of the sites but also the order of the input and output states, and thus the local gate is transformed as $\mathcal{P}_{\Phi} \to P \circ \mathcal{P}_{\Phi} \circ P$, where $P$ is the Swap operation.

The correlations in (18) can be expressed in terms of two different one-site transfer operators. In particular, we define linear maps $\mathcal{F}_{\pm} : L^2(M) \to L^2(M)$, where $\pm$ corresponds to even/odd parity respectively. Graphically the transfer operators are represented in Fig. 4

Here one can also observe the reflection property mentioned above, which maps the transfer operator of one chirality to the other. For this reason, from now on we are going to omit the label $\pm$ and focus only on the right moving light-cone edge with $\mathcal{F}_{+} \equiv \mathcal{F}$. Afterwards, according to (18), the correlations along the edges of the light cone take the form

$$C_{ab}(2t; t) = \langle \circ | \hat{b} \; \mathcal{F}^{2t} \; \hat{a} | \circ \rangle - \langle \circ | \hat{b} | \circ \rangle \langle \circ | \hat{a} | \circ \rangle. \tag{19}$$

This is an important exact result, which shows that in dual-symplectic circuits, the correlations are determined explicitly by transfer operators acting on a single site. The operator $\mathcal{F}$ is in general not Hermitian, but as proven in the Appendix B, it is positive, and a weak contraction. Assuming that it has a pure point spectrum, as will be the case in the spin chain examples studied below, its spectral decomposition reads

$$\mathcal{F} = \sum_{i=0}^{\infty} \mu_i |\mu_i^R\rangle \langle \mu_i^L|, \tag{20}$$

where we indicated the left and right eigenvectors as $|\mu_i^R\rangle$, $\langle \mu_i^L|$, and ordered the eigenvalues as $|\mu_0| \geq |\mu_1| \geq \ldots$. As it is a weak contraction, its spectrum lies in the unit disk, $|\mu_i| \leq 1$. We

Table 1: The table with all the different levels of ergodicity, with respect to the non-trivial eigenvalues.

| eigenvalues $\mu_i, \quad i \neq 0$ | non−interacting | non−ergodic | ergodic, non−mixing | ergodic, mixing |
|---|---|---|---|---|
| $\mu_i = 1$ | | $\mu_1, \mu_2, ..., \mu_{j<\infty}$ | | |
| $|\mu_i| = 1$ | $\mu_1, \mu_2, ..., \mu_\infty$ | $\mu_{j+1}, \mu_{j+2}, ..., \mu_{j+m<\infty}$ | $\mu_1, \mu_2, ..., \mu_{j<\infty}$ | |
| $|\mu_i| < 1$ | | $\mu_{j+m+1}, \mu_{j+m+2}, ..., \mu_\infty$ | $\mu_{j+1}, \mu_{j+2}, ..., \mu_\infty$ | $\mu_1, \mu_2, ..., \mu_\infty$ |

should also note, that as proved in [32], the eigenvalues with $|\mu_i| = 1$ have equal algebraic and geometric multiplicity, and thus their Jordan blocks are trivial. A direct consequence of the dual-symplectic nature of $\mathcal{P}_\Phi$, is that the uniform measure is invariant under the action of $\mathcal{F}$. Therefore, the transfer operator always has the trivial eigenvalue $\mu_0 = 1$ with $|\mu_0^R\rangle = |\circ\rangle$ and $\langle\mu_0^L| = \langle\circ|$. Plugging the spectral decomposition (20) in Eq. (19), we obtain:

$$C_{ab}(v_c t; t) = \sum_{i=1}^{\infty} \langle\circ|\hat{b}|\mu_i^R\rangle \langle\mu_i^L|\hat{a}|\circ\rangle \mu_i^{2t}, \tag{21}$$

where the $i = 0$ term in the sum cancels with the second term in Eq. (19).

Note that the spectrum of $\mathcal{F}$ can be used to analyse the ergodicity of single-site observables. Depending on how many non-trivial eigenvalues are equal to 1 or have a unit modulus, dual-symplectic circuits can demonstrate different levels of ergodicity, as we show in Table 1. In particular, in the non-interacting case, all eigenvalues are unimodular with $|\mu_i| = 1$, and all correlations either remain constant or oscillate around zero, and in the non-ergodic case, where more than one but not all eigenvalues are equal to 1, the correlations decay to a non-thermal value. When the system is ergodic and non-mixing, all non-trivial $\mu_i$ are not equal to 1, and at least one of them has unit modulus leading to correlations which oscillate around zero, and thus their time averages vanish at long times. Finally, for an ergodic and mixing system, all $\mu_i$ are within the unit disk and all correlations decay to zero. A general example for the non-interacting case is the dual-symplectic local gate $\mathcal{P}_\Phi = P \circ (\mathcal{P}_{\phi_1} \otimes \mathcal{P}_{\phi_2})$ with $P$ being the Swap gate and $\phi_1, \phi_2$ being single-site symplectic maps.

## 4 The Ising swap model

Previously, we were studying an abstract dual-symplectic circuit. In order to test our general analytical results, we now focus on a 1D classical spin chain, where the local phase space is the unit sphere $M \equiv \mathcal{S}^2$. We denote the coordinates as $\vec{X}_i \equiv \vec{S}_i$ with the constraint $|\vec{S}_i| = 1$, and introduce the 3-parameter family of dual-symplectic local gates, which read

$$\Phi_{(\alpha,\beta,\gamma)} := \left(R_x(\beta) \otimes R_x(\gamma)\right) \circ I_\alpha \circ \left(R_x(\gamma) \otimes R_x(\beta)\right). \tag{22}$$

Here, the operation $R_n(\theta), \theta \in [0, 2\pi)$ denotes a single spin rotation – SO(3) rotation matrix – with respect to axis $n \in \{x, y, z\}$ by the angle $\theta$. We denote with $I_\alpha$, the Ising Swap gate, whose action on a pair of sites reads

$$I_\alpha(\vec{S}_1, \vec{S}_2) = \left(R_z(\alpha S_1^z)\vec{S}_2, R_z(\alpha S_2^z)\vec{S}_1\right), \tag{23}$$

with $\alpha$ being the coupling constant of the interactions, $R_z(\theta)$ is a rotation around the z-axis and $S_i^z$ is the z-component of $\vec{S}_i$. Assuming the SO(3) Poisson bracket on the unit sphere

$$\{S_i^a, S_j^b\} = \delta_{ij}\epsilon_{abc}S_i^c, \tag{24}$$

with $\epsilon_{abc}$ being the Levi-Civita symbol, it is easy to recognize Eq. (23) as the symplectic evolution of two sites under the Hamiltonian $H_{12} = \alpha S_1^z S_2^z$ for a time step $\delta t = 1$, followed by a Swap operation $(S_1^n, S_2^n) \rightarrow (S_2^n, S_1^n)$. The spin variables, as we can see from (24), are not the pairs $(q, p)$ of conjugate variables, as would be expected in symplectic dynamics. In general, there is not a unique choice of conjugate variables since a symplectic transformation maps from one set of conjugate variables to another. However, here we choose the pairs $\varphi_i, z_i$ with $z_i$ being the cartesian coordinate along the z-axis and $\varphi_i$ the azimuthal angle of the $i$-th site and so they satisfy

$$\{\varphi_i, z_j\} = \delta_{ij}, \quad \{\varphi_i, \varphi_j\} = \{z_i, z_j\} = 0. \tag{25}$$

The spin variables are just vectors of the unit sphere, namely, they are related to $\varphi_i, z_i$ in the following way

$$S_i^x = \sqrt{1 - z_i^2}\cos(\varphi_i), \qquad S_i^y = \sqrt{1 - z_i^2}\sin(\varphi_i), \qquad S_i^z = z_i, \tag{26}$$

and one can check that (26) satisfies the SO(3) Poisson bracket (24).

We explicitly demonstrate in Appendix C, that (22) satisfies (15), allowing us for equivalent interpretations of the diagrams in both time and space directions. Following the same method as employed in Appendix A one finds that the space-time dual of our model is defined as

$$\tilde{\Phi}_{(\alpha,\beta,\gamma)} \equiv (\mathbb{1} \otimes (-\mathbb{1})) \circ \Phi_{(\alpha,-\beta,\gamma)} \circ ((-\mathbb{1}) \otimes \mathbb{1}), \tag{27}$$

where we indicated by $\mathbb{1}$ the identity map and by $-\mathbb{1}$ the change of sign of all components $S_i^a \rightarrow -S_i^a$. Thus the dual dynamics differs from the temporal one, by a simple sign-gauge transformation. As described in [15], our map $\Phi_{(\alpha,\beta,\gamma)}$ is space-time self-dual because flipping of the spins in a checkerboard pattern recovers spatial dynamics from the temporal one. Dual-symplectic circuits with local gates (22) accommodate both ergodic and integrable cases depending on the choice of parameters. For example, for $\alpha = 0$ the model becomes a trivial non-interacting one and so integrability is expected. Another integrable case is when both $\beta, \gamma$ take either of the values $0, \pi$, where the dynamics preserve the $z$-components of the spins along their respective light rays leading to conserved extensive quantities along the parity's bi-partition of the lattice. This type of local conserved quantities are called gliders and have been previously studied in dual-unitary quantum circuits [33]. Later, in (33) we provide analytical results for the auto-correlation of the z-components at integrable points when they do not decay to zero. These models are also known as super-integrable as they support an exponentially large number of extensive conserved quantities, which can be obtained by summing arbitrary products of $z$-components along the aforementioned bipartitions, e.g. $\mathcal{Q} = \sum_i z_i z_{i+2} z_{i+4}$. At the integrable points of parameter space, the trajectories in phase space are bounded on invariant tori and the Lyapunov spectrum vanishes [34], whereas away from those points chaotic behaviour is expected to arise. In Fig. 5, we present some examples of the Lyapunov spectrum at chaotic points of our Ising Swap model, where it demonstrates a positive maximal Lyapunov exponent and thus sensitivity to initial conditions, which is a characteristic property of a chaotic system.

Having chosen, our family of local gates we proceed with the calculation of the correlations. As explained in the general formalism of Sec. 3.2, this requires the calculation of the transfer operators $\mathcal{F}$, acting on the single-site functions. In Appendix D, we present an analytical calculation of the transfer operator, in both the phase space and the density space:

$$f = R_x(\gamma) Q(\alpha) R_x(\gamma), \qquad \mathcal{F} \equiv \mathcal{P}_f, \tag{28}$$

with $Q(\alpha) = \frac{1}{2}\int_{-1}^{1} dz' R_z(\alpha z')$ and $f : M \rightarrow M$. The transfer operator is the Frobenius-Perron operator of $f$, and its kernel is given in the same way as in (7), for a single site phase space.

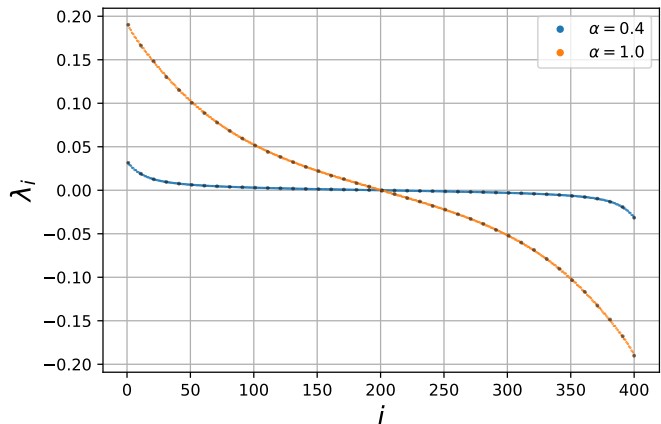

Figure 5: Lyapunov spectrum $\lambda_i$ of the Ising Swap model, for two different coupling constants $\alpha = 0.4, 1$, angles $\beta = \sqrt{2}\pi, \gamma = \sqrt{3}\pi/2$ and system size $N = 200$. The figures were obtained for $t = 800$ and a sample size of $N_{sample} = 10^4$ initial states drawn from the uniform measure. The black circles represent the Lyapunov spectrum at every 10 exponents at time $t = 700$ showing an excellent time convergence for $\lambda_i$. The spectrum is symmetric with respect to the horizontal axis, as expected for a symplectic system and has a positive maximal Lyapunov exponent indicating chaoticity for $\Phi_{\alpha,\beta,\gamma}$.

Rotations preserve the total angular momentum and since, $\mathcal{F}$, according to (28), is a composition of rotations it shares the same property, as shown in the Appendix D. More explicitly, we denote as $J_i, i = x, y, z$ the generators of single-site rotations, and $J^2 = \sum_i J_i^2$ as the angular-momentum-squared which satisfies $[J^2, J_i] = 0, \forall i$ and thus, commutes with every rotation operation. Therefore, $\mathcal{F}$ commutes with the total angular momentum operator and thus, has a block-diagonal form in its eigenvalues, as we demonstrate in Appendix D. However, this is not a consequence of an underlying rotational symmetry but rather of the specific form of the local gate $\mathcal{P}_{\Phi_{\alpha,\beta,\gamma}}$. Indeed, the Ising swap gate in $\mathcal{P}_{\Phi_{\alpha,\beta,\gamma}}$ involves a *non-linear rotation*, i.e. a rotation whose angle depends on the $z$ component of the neighbouring spin. Because of this nonlinearity, it is not block- diagonal with respect to the eigenvalues of $J^2$, as we show in Appendix E. Nonetheless, in going from the local gate $\mathcal{P}_{\Phi_{\alpha,\beta,\gamma}}$ to the transfer operator $\mathcal{F}$, the neighbouring site is, by definition Fig. 4, in the equilibrium state, so that its $z$ component can be integrated over, thus leading to the operator $Q(\alpha)$, which is a linear superposition of rotations.

It is worth noting, that (28) is completely independent of $\beta$. The correlation with the opposite chirality is simply recovered from the middle point reflection $P \circ \mathcal{P}_{\Phi_{\alpha,\beta,\gamma}} \circ P = \mathcal{P}_{\Phi_{\alpha,\gamma,\beta}}$, which is equivalent with changing $\beta, \gamma \to \gamma, \beta$.

The fact that (28),(22) are expressed in terms of rotations, suggests the use of spherical harmonics as a convenient basis for the $L^2$ density space, and indeed in Appendices D, E, we obtain analytical expressions for their representations on this basis. We choose the conjugate variables $z, \varphi$ for the parametrization of $\mathcal{S}^2$. Then, the spherical harmonics $|\ell, m\rangle \to Y_{\ell,m}(z, \varphi)$ for $\ell = 0, 1, \ldots$ and $|m| \leq \ell$ form a suitable orthonormal basis for $L^2$ functions. Our approach is based on finding the representation of the transfer operator on this basis. As we already mentioned, the transfer operator $\mathcal{F}$ preserves the total angular momentum and thus has a block diagonal form in $\ell$, with each block having dimension $2\ell + 1$. It follows, that the eigenvectors and eigenvalues in Eq. (20), can be indexed by a block index $\ell$ and an index $\tilde{m}$ within each

block. Thus, Eq. (21) assumes the form

$$C_{a,b}(2t,t) = \sum_{\ell=1}^{\infty} \sum_{\tilde{m}=-\ell}^{\ell} \langle \circ | \hat{b} | \mu_{\ell,\tilde{m}}^R \rangle \langle \mu_{\ell,\tilde{m}}^L | \hat{a} | \circ \rangle \, \mu_{\ell,\tilde{m}}^{2t} \,. \tag{29}$$

From this expression, it follows that if the local observables $|a_x\rangle, |b_y\rangle$ lie within a finite number of total angular momentum subspaces, then the sum in Eq. (29) contains a finite number of terms. In particular, only the common values of $\ell$ between the two observables will matter. For example, the observable $a(z,\phi) = z^2$ has non-vanishing overlaps, only for $\ell = 0, 2$. Similarly, any polynomial in the variable $z$ involves a finite number of blocks. This is an important property of our system, proved in Appendix F, since it suggests that a finite set of exponentials (in $t$) fully captures the behaviour of the 2-point correlations, whenever one of the two observables $a, b$ involves only a finite number of $\ell$ blocks. In practice, one can calculate the exact dynamical correlations, by diagonalizing the relevant finite-dimensional blocks of $\mathcal{F}$. Moreover, observables which have no such overlapping subspaces lead to vanishing correlations for every $t$.

At this point, we provide some analytical results for the choice of $a(z,\phi) = z^n$, $b(z,\phi) = z$ with $n \in \mathbb{Z}^+$. In this case, $a$ has non vanishing overlaps for $\ell = 0, 2, \ldots n$, if $n$ even and $\ell = 1, 3, \ldots n$, if $n$ odd and $b$ for $\ell = 1$. We can see that for the case of $n$ even, there are no common overlapping subspaces between the two observables, and thus the correlations vanish for all $t$. However, when $n$ is odd, the correlations depend only on the $\ell = 1$ block of $\mathcal{F}$, and by using (28) one can explicitly find, that the eigenvalues of this block are the following

$$\begin{aligned}
\mu_{1,0} &= \frac{\sin(\alpha)}{\alpha} \,, \\
\mu_{1,-1} &= \frac{(\alpha + \sin(\alpha))\cos(2\gamma) - \Delta(\alpha,\gamma)}{2\alpha} \,, \\
\mu_{1,1} &= \frac{(\alpha + \sin(\alpha))\cos(2\gamma) + \Delta(\alpha,\gamma)}{2\alpha} \,,
\end{aligned} \tag{30}$$

where $\Delta(\alpha,\gamma) = \sqrt{(\alpha + \sin\alpha)^2 \cos^2(2\gamma) - 4\alpha\sin(\alpha)}$. Since only the $\ell = 1$ subspace contributes we only need the overlaps of the observables with this subspace,

$$\langle 1m | z^n \rangle = \frac{2\sqrt{3\pi}}{n+2} \delta_{m,0} \,. \tag{31}$$

The observable $z^n$ does not depend on the azimuthal angle, and therefore it depends only on the spherical harmonics $|\ell m\rangle$ with $m = 0$. By diagonalizing the block of $\mathcal{F}$, that corresponds to $\ell = 1$ and using (30),(31), one can recover the exact expression for the correlations:

$$\begin{aligned}
C_{a,b}(2t,t) &= \frac{1}{2^{2t+1}(n+2)} \left( \mathcal{E}^+(t) + \mathcal{E}^-(t) \right) , \\
\mathcal{E}^\pm(t) &= \left( 1 \pm \frac{\alpha - \sin\alpha}{\Delta} \right) \left( \frac{(\alpha + \sin\alpha)\cos(2\gamma) \pm \Delta}{\alpha} \right)^{2t} ,
\end{aligned} \tag{32}$$

where the other chirality is recovered with $\gamma \to \beta$. In the special integrable cases, one finds

$$\lim_{\alpha \to 0} C_{a,b}(2t,t) = \frac{\cos(4\gamma t)}{2+n} \,, \qquad \lim_{\gamma \to 0} C_{a,b}(2t,t) = \frac{1}{2+n} \,. \tag{33}$$

In Fig. 6, we present the results of the numerics, which show, that the correlations survive only along the edges of the causal cone, and verify (32) in the case of $n = 1$. Here it is important to note, that we make use of the symmetries of our circuit for the numerical evaluation of the correlations. In particular, apart from the 2-site translation invariance, there is also a 1 time step translation invariance, because the correlations are evaluated over the invariant measure, and both of these symmetries allow us to perform averaging, over a larger sample size and obtain more accuracy for the numerical data.

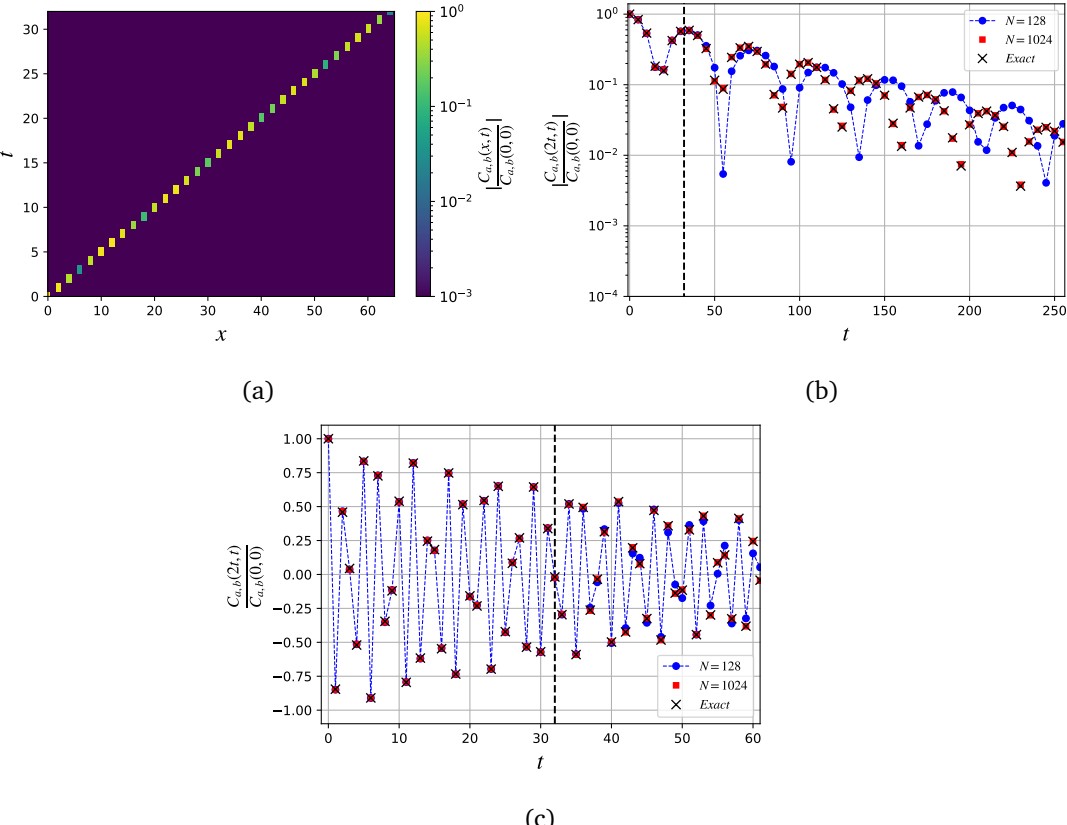

Figure 6: Auto-correlations for the $S^z$ spin component, normalized by the maximum value $C_{a,b}(0,0)$ for systems with $N = 128, 1024$ spins, the parameters $\alpha = 0.3$ and $\beta = \frac{\sqrt{2}}{4}\pi, \gamma = \frac{\sqrt{2}}{2}\pi$, and with a sampling size of $N_{sample} = 5 \times 10^4$ for the initial conditions. (a): The space-time correlator $|C_{a,b}(x,t)/C_{a,b}(0,0)|$ for $N = 128$ where we can observe that it vanishes away from the edge ($x = v_c t$) of the causal cone. (b): The comparison of the theoretical result for $|C_{a,b}(x,t)/C_{a,b}(0,0)|$, obtained from (32), on the right edge of the causal cone with exact diagonalization for $\mathcal{F}$ in the $\ell = 1$ subspace. The numerical results are obtained for two different system sizes $N = 128, 1024$ and shown with the time step of 5 on a log scale. The dashed line represents the time moment $t = N/4$ for the system of length $N = 128$, where our theory (which gives the results in the thermodynamic limit) no longer applies. The numerical results for the system with length $N = 128$, no longer agree with the exact results after this time moment, but a larger system $N = 1024$ demonstrates excellent agreement with the theory at longer times. (c): The comparison of $C_{a,b}(x,t)/C_{a,b}(0,0)$ on the right edge of the causal cone, shown using a linear scale for the vertical axis.

## 5 Conclusion

We have provided an exact approach for the calculation of the dynamical correlation functions in dual-symplectic classical circuits, showing that the correlations do not vanish only along the edges of the light cones, and are completely specified in terms of a weakly contracting and positive single-site transfer operator. We would like to stress, that our method is valid not only for dual-symplectic systems, as it is easy to check that any local gate $\Phi$ which is volume preserving and which also has a volume-preserving dual map $\tilde{\Phi}$ and satisfies (15), satisfies

also (16) and exhibits the same diagrammatic behaviour. Every symplectic map is volume and orientation-preserving, but the group of symplectic diffeomorphisms is significantly smaller than that of the volume-preserving ones (Non-squeezing theorem [35]). Consequently, there is a larger set of dynamical ergodic systems, which exhibit our diagrammatic representation, having correlations, which vanish everywhere except the edges of the light-cone. In addition, we prove that for the important case of the Ising Swap model, the transfer operator exhibits a block-diagonal form, which leads to an expansion, involving only the common $\ell$ subspaces of the observables. This property has a great advantage, as truncation is not required, and one can obtain analytical results using exact diagonalization within each (finite) block.

We close with some naturally arising questions. Is it possible to find a more general characterization or complete parametrization of the dual symplectic circuits which may help in, also finding a parametrization of dual-unitary gates [32] to larger than qubits local spaces? Could one find exact results for other initial densities as has already been demonstrated in the dual unitary case [31]? Our formalism can be a stepping stone to studying these types of questions for the novel class of dual-symplectic systems.

## Acknowledgments

We acknowledge fruitful discussions with B. Bertini and Ž. Krajnik which, in particular, helped us to understand the role of the Jacobian when transforming between the time and space directions.

**Funding information** T.P acknowledges financial support from Program P1-0402, and Grants N1-0219 and N1-0233 of the Slovenian Research and Innovation Agency (ARIS). D.K acknowledges partial support from Labex MME-DII (Modèles Mathématiques et Economiques de la Dynamique, de l'Incertitude et des Interactions), ANR11-LBX-0023 and LPTM ANR INEX 2020 EMERGENCE CMTNEQ grants. A. D. L. acknowledges support by the ANR JCJC Grant ANR-21- CE47-0003 (TamEnt). A.C acknowledges support from the EUTOPIA PhD Co-tutelle program.

## A   Dual picture and the change of variables

In this Appendix, we obtain analytically the extra condition, that comes from demanding the diagrammatic representation to be equivalent in both the original and the dual picture. We start with the simple case of two sites, and thus we are working in the phase space $M_2$. Assuming two arbitrary scalars $A, B \in L^2(M_2)$, we are interested in the Hermitian product $\langle B | \mathcal{P}_\Phi | A \rangle$, which corresponds to the following diagram,

$$\langle B | \mathcal{P}_\Phi | A \rangle = \quad \text{(A.1)}$$

With the help of (6),(7) on can write (A.1) as an integral

$$\langle B|\mathcal{P}_\Phi|A\rangle = \int d\vec{X}_1 d\vec{X}_2 d\vec{X}_1' d\vec{X}_2' \, A(\vec{X}_1,\vec{X}_2) B^*(\vec{X}_1',\vec{X}_2') \delta\big((\vec{X}_1',\vec{X}_2')-\Phi(\vec{X}_1,\vec{X}_2)\big), \qquad (A.2)$$

in the time direction. If the same diagram can be interpreted equivalently in the dual picture one should obtain that

$$\langle B|\mathcal{P}_\Phi|A\rangle = \int d\vec{X}_1 d\vec{X}_2 d\vec{X}_1' d\vec{X}_2' \, A(\vec{X}_1,\vec{X}_2) B^*(\vec{X}_1',\vec{X}_2') \delta\big((\vec{X}_2,\vec{X}_2')-\tilde{\Phi}(\vec{X}_1,\vec{X}_1')\big), \qquad (A.3)$$

where we used the definition of the dual map, as presented in Fig. 3, and exchanged the input from being the local states in space $(\vec{X}_1,\vec{X}_2)$ to local states in time $(\vec{X}_1,\vec{X}_1')$. In order for both (A.2),(A.3) to be valid one has to demand that

$$\delta\big((\vec{X}_1',\vec{X}_2')-\Phi(\vec{X}_1,\vec{X}_2)\big) = \delta\big((\vec{X}_2,\vec{X}_2')-\tilde{\Phi}(\vec{X}_1,\vec{X}_1')\big). \qquad (A.4)$$

The equivalence of the delta functions implies the equivalence of the change of variables from one picture to the other and thus imposes a restriction on the Jacobian of this transformation. In particular, we define $g(\vec{X}_1,\vec{X}_2,\vec{X}_1',\vec{X}_2') = (\vec{X}_1',\vec{X}_2')-\Phi(\vec{X}_1,\vec{X}_2)$ and assume that we want to change variables with respect to $\vec{X}_2,\vec{X}_2'$. Then from (A.4), one obtains

$$\frac{\delta\big(g(\vec{X}_1,\vec{X}_2,\vec{X}_1',\vec{X}_2')=0\big)}{|\det(Dg)|} = \delta\big((\vec{X}_2,\vec{X}_2')-\tilde{\Phi}(\vec{X}_1,\vec{X}_1')\big), \qquad (A.5)$$

where $Dg$ is the Jacobian matrix of the transformation $g$ with respect to $\vec{X}_2,\vec{X}_2'$, and it is found as

$$Dg = \begin{bmatrix} -\dfrac{\partial \Phi^1(\vec{X}_1,\vec{X}_2)}{\partial \vec{X}_2} & 0 \\[2mm] -\dfrac{\partial \Phi^2(\vec{X}_1,\vec{X}_2)}{\partial \vec{X}_2} & \mathbb{1} \end{bmatrix}, \qquad (A.6)$$

where we decomposed $\Phi$ into single-site outputs

$$\Phi(\vec{X}_1,\vec{X}_2) = \big(\Phi^1(\vec{X}_1,\vec{X}_2), \Phi^2(\vec{X}_1,\vec{X}_2)\big). \qquad (A.7)$$

The solutions of $g(\vec{X}_1,\vec{X}_2,\vec{X}_1',\vec{X}_2') = 0$ with respect to $\vec{X}_1,\vec{X}_1'$, are the points of the dual map $(\vec{X}_2,\vec{X}_2') = \tilde{\Phi}(\vec{X}_1,\vec{X}_1')$, and if we assume that $\Phi$ has a unique and bijective dual map then from (A.5) we obtain that

$$\frac{\delta\big((\vec{X}_2,\vec{X}_2')-\tilde{\Phi}(\vec{X}_1,\vec{X}_1')\big)}{|\det(Dg)|} = \delta\big((\vec{X}_2,\vec{X}_2')-\tilde{\Phi}(\vec{X}_1,\vec{X}_1')\big). \qquad (A.8)$$

This is true in $M_2$ if $|\det(Dg)| = 1$, which according to (A.6) leads to

$$\left|\det\left(\frac{\partial \Phi^1(\vec{X}_1,\vec{X}_2)}{\partial \vec{X}_2}\right)\right| = 1, \quad \forall \vec{X}_1,\vec{X}_2 \in M \times M. \qquad (A.9)$$

This is a necessary condition that the local gate $\Phi$ has to satisfy, in order for the diagrammatics to be equivalent in both the time picture and the dual picture.

We continue by showing that, as expected, (A.9) implies a respective condition for $\tilde{\Phi}$, meaning that, we get an equivalent result, even when the change of variables happens from the dual picture back to the original one. Firstly, if we denote the output of as $\Phi(\vec{X}_1,\vec{X}_2) = (\vec{X}_1',\vec{X}_2')$, from (A.7) we obtain:

$$\vec{X}_1' = \Phi^1(\vec{X}_1,\vec{X}_2), \qquad (A.10)$$

$$\vec{X}_2' = \Phi^2(\vec{X}_1,\vec{X}_2). \qquad (A.11)$$

Since, by the definition $(\vec{X}_2, \vec{X}'_2) = \tilde{\Phi}(\vec{X}_1, \vec{X}'_1)$ we need to find $\vec{X}_2, \vec{X}'_2$ as functions of $\vec{X}_1, \vec{X}'_1$ and thus we invert (A.10) with respect to $\vec{X}_2$ and replace it in (A.11),

$$\vec{X}_2 = (\Phi^1_{\vec{X}_1})^{-1}(\vec{X}'_1),$$
$$\vec{X}'_2 = \Phi^2\big(\vec{X}_1, (\Phi^1_{\vec{X}_1})^{-1}(\vec{X}'_1)\big),$$

where we assume that the first output $\Phi^1(\vec{X}_1, \vec{X}_2) = \Phi^1_{\vec{X}_1}(\vec{X}_2)$ is a family of invertible maps $\Phi^1_{\vec{X}_1} : M \to M$ for every $\vec{X}_1 \in M$. Finally the dual map of $\Phi$ reads

$$\tilde{\Phi}(\vec{X}_1, \vec{X}'_1) = \Big((\Phi^1_{\vec{X}_1})^{-1}(\vec{X}'_1), \Phi^2\big(\vec{X}_1, (\Phi^1_{\vec{X}_1})^{-1}(\vec{X}'_1)\big)\Big). \tag{A.12}$$

Then, from (A.9), one can deduce that the inverse of $\Phi^1_{\vec{X}_1}(\vec{X}_2)$ has also a Jacobian equal to one,

$$\left| \det\left( \frac{\partial (\Phi^1_{\vec{X}_1})^{-1}(\vec{X}'_1)}{\partial \vec{X}'_1} \right) \right| = 1, \tag{A.13}$$

which is exactly the respective condition (A.9) for the dual map $\tilde{\Phi}$, and thus making this condition consistent with the the dual operation, being an involution.

One can follow the same procedure for the case when the diagrams are being interpreted from right to left and thus, the change of variables we need to perform in (A.4) is with respect to $\vec{X}'_1, \vec{X}_1$. This is equivalent to defining a dual map as in Fig. 3, but with the swapping of the other diagonal of the legs of $\Phi$. This dual map is the solution of $g(\vec{X}_1, \vec{X}_2, \vec{X}'_1, \vec{X}'_2) = 0$ with respect to $\vec{X}'_1, \vec{X}_1$. Similarly, we demand that $\Phi^2(\vec{X}_1, \vec{X}_2) = \Phi^2_{\vec{X}_2}(\vec{X}_1)$ is a family of invertible maps $\Phi^2_{\vec{X}_2} : M \to M$ for every $\vec{X}_2 \in M$, and we obtain the extra condition

$$\left| \det\left( \frac{\partial \Phi^2(\vec{X}_1, \vec{X}_2)}{\partial \vec{X}_1} \right) \right| = 1, \quad \forall \vec{X}_1, \vec{X}_2 \in M \times M. \tag{A.14}$$

In addition, one can prove, in the same manner as for (A.13), that (A.14) is consistent with the dual map being an involution.

## B  Weak contractivity and positivity of $\mathcal{F}_\pm$

In this Appendix, we show, that the single-site transfer operator $\mathcal{F} (\equiv \mathcal{F}_+)$ is a weak contraction, as well as being a positive operator. First, as mentioned in the main text, the map $\mathcal{P}_\Phi$ is unitary in $L^2(M \times M)$ since it preserves the $L^2$-norm. Then according to this, from (D.1) we can obtain for every $\rho_1, \rho_2 \in L^2(M)$:

$$|\langle \rho_1 | \mathcal{F} | \rho_2 \rangle| = |((\langle \circ | \otimes \langle \rho_1 |) \, \mathcal{P}_\Phi \, (|\rho_2\rangle \otimes |\circ\rangle))| \leq \| \, |\circ\rangle \otimes |\rho_1\rangle \|_2 \, \|\mathcal{P}_\Phi \, (|\rho_2\rangle \otimes |\circ\rangle)) \|_2$$
$$= \| \, |\circ\rangle \otimes |\rho_1\rangle \|_2 \, \| \, |\rho_2\rangle \otimes |\circ\rangle \|_2 = \|\rho_1\|_2 \, \|\rho_2\|_2, \tag{B.1}$$

where we used the Cauchy-Schwarz inequality and the fact, that the state $|\circ\rangle$ is normalised. Now by setting $|\rho_1\rangle = \mathcal{F}|\rho_2\rangle$, one recovers

$$\|\mathcal{F}|\rho_2\rangle\|_2 \leq \|\rho_2\|_2, \tag{B.2}$$

for every $\rho_2 \in L^2(M)$. This suggests that the single-site transfer operator is a weak contraction.

The positivity is a direct consequence of the properties of the Frobenius-Perron operator. In particular, let us assume that $\rho \in L^2(M)$ and $\vec{X} \in M$. Then we are interested in the value of the scalar $\langle \vec{X} | \mathcal{F} | \rho \rangle$. It is sufficient to prove, that it is always positive if $\rho \geq 0$. The calculation is based on the use of Eq. (D.1), from which we get

$$\langle \vec{X} | \mathcal{F} | \rho \rangle = \left( \langle \circ | \otimes \langle \vec{X} | \right) \mathcal{P}_\Phi \left( | \rho \rangle \otimes | \circ \rangle \right). \tag{B.3}$$

The scalar $| \circ \rangle \rightarrow u_\circ(\vec{X}) = 1/\sqrt{|M|}$ is positive, thus if we assume $\rho \geq 0$ then $|\rho\rangle \otimes |\circ\rangle \rightarrow (\rho u_\circ)(\vec{X}_1, \vec{X}_2) = \rho(\vec{X}_1)u_\circ(\vec{X}_2)$ is also a non-negative scalar. Now as the last step, we need to note that $\mathcal{P}_\Phi$ is a Frobenius-Perron operator, which by definition is positive and thus implies that $\mathcal{P}_\Phi \left( |\rho\rangle \otimes |\circ\rangle \right) \geq 0$. As a consequence the value (B.3), is non-negative for every $\vec{X} \in M$ meaning that

$$\mathcal{F}|\rho\rangle \geq 0, \quad \text{for any} \quad \rho \geq 0 \in L^2(M).$$

In the same way, one can prove these properties for $\mathcal{F}_-$ as well.

## C  Diagrammatic equivalence's conditions for $\Phi_{\alpha,\beta,\gamma}$

In this part, we will show, that our Ising Swap model satisfies Eq. (15) for classical spin variables in $\mathcal{S}^2$. We start by decomposing (22) into single site components $\Phi_{\alpha,\beta,\gamma}(\vec{S}_1, \vec{S}_2) = \left( \Phi^1_{\alpha,\beta,\gamma}(\vec{S}_1, \vec{S}_2), \Phi^2_{\alpha,\beta,\gamma}(\vec{S}_1, \vec{S}_2) \right) = (\vec{S}'_1, \vec{S}'_2)$ with

$$\vec{S}'_1 = R_x(\beta) R_z\left( \alpha(R_x(\gamma)\vec{S}_1)^z \right) R_x(\beta) \vec{S}_2, \tag{C.1}$$

$$\vec{S}'_2 = R_x(\gamma) R_z\left( \alpha(R_x(\beta)\vec{S}_2)^z \right) R_x(\gamma) \vec{S}_1. \tag{C.2}$$

First, we observe that, $\Phi^1_{\alpha,\beta,\gamma}(\vec{S}_1, \vec{S}_2) = R_x(\beta) R_z\left( \alpha(R_x(\gamma)\vec{S}_1)^z \right) R_x(\beta) \vec{S}_2$ and $\Phi^2_{\alpha,\beta,\gamma}(\vec{S}_1, \vec{S}_2) = R_x(\gamma) R_z\left( \alpha(R_x(\beta)\vec{S}_2)^z \right) R_x(\gamma) \vec{S}_1$, which makes its Jacobian matrix over $\vec{S}_2$ and $\vec{S}_1$ respectively, a composition of rotations and implies that

$$\left| \det\left( \frac{\partial \Phi^1_{\alpha,\beta,\gamma}(\vec{S}_1, \vec{S}_2)}{\partial \vec{S}_2} \right) \right| = \left| \det\left( R_x(\beta) R_z\left( \alpha(R_x(\gamma)\vec{S}_1)^z \right) R_x(\beta) \right) \right| = 1, \quad \forall \vec{S}_1, \vec{S}_2 \in \mathcal{S}^2 \times \mathcal{S}^2,$$

$$\left| \det\left( \frac{\partial \Phi^2_{\alpha,\beta,\gamma}(\vec{S}_1, \vec{S}_2)}{\partial \vec{S}_1} \right) \right| = \left| \left( R_x(\gamma) R_z\left( \alpha(R_x(\beta)\vec{S}_2)^z \right) R_x(\gamma) \right) \right| = 1, \quad \forall \vec{S}_1, \vec{S}_2 \in \mathcal{S}^2 \times \mathcal{S}^2.$$

## D  Block-diagonal form of the operator $\mathcal{F}_\pm$

In this Appendix, we calculate the matrix of the one-site transfer operator in terms of spherical harmonics and prove that it has a block-diagonal form in $\ell$. Our calculation is based on the interpretation of Fig. 4. In particular, for a general local gate, one can interpret $\mathcal{F}_\pm$ either in the time direction, where the dynamics are performed by $\Phi$, or in the space direction, where we can use the dual picture with $\tilde{\Phi}$. Both of these pictures are equivalent, but here we choose the former one. As in the main text, we focus on the right-moving chirality $\mathcal{F}_+ \equiv \mathcal{F}$ and omit the $\pm$ label. According to this choice, one can see from Fig. 4, that the transition amplitudes of $\mathcal{F}$ for two arbitrary densities (functions) $\rho_1, \rho_2$ from $L^2(M)$ are the following

$$\langle \rho_1 | \mathcal{F} | \rho_2 \rangle = \left( \langle \circ | \otimes \langle \rho_1 | \right) \mathcal{P}_\Phi \left( | \rho_2 \rangle \otimes | \circ \rangle \right). \tag{D.1}$$

We note that this holds for any dual-symplectic gate. We focus on the Ising Swap model, where since $|\circ\rangle = |00\rangle$ in the basis spherical harmonics one can use (D.1),(E.6),(E.7) to obtain

$$\langle \ell m | \mathcal{F} | \ell' m' \rangle = \langle 00, \ell m | \mathcal{P}_{\Phi_{\alpha,\beta,\gamma}} | \ell' m', 00 \rangle = \delta_{\ell,\ell'} \sum_{q_2=-\ell}^{\ell} \langle \ell m | \mathcal{P}_{R_x(\gamma)} | \ell q_2 \rangle \frac{\sin(\alpha q_2)}{a q_2} \langle \ell q_2 | \mathcal{P}_{R_x(\gamma)} | \ell m' \rangle,$$
(D.2)

where we used that $T(0) = \mathbb{1}$, $C_{0,0,0}^{0,0,0} = 1$ and $j_0(x) = \sin(x)/x$, and the fact that a constant scalar is invariant under rotations, thus $\langle 00 | \mathcal{P}_{R_x(\beta)} | 00 \rangle = 1$. This expression can be further simplified by defining the map $Q(\alpha) : M \to M$

$$Q(\alpha) = \frac{1}{2} \int_{-1}^{1} dz' R_z(\alpha z').$$
(D.3)

The spherical harmonics form the eigenbasis of the Frobenius-Perron operator of rotations around the $z-$axis and in particular $\langle \ell_1 m_1 | \mathcal{P}_{R_z(\theta)} | \ell_2 m_2 \rangle = e^{-i m_1 \theta} \delta_{\ell_1,\ell_2} \delta_{m_1,m_2}$ for a rotation of an angle $\theta \in [0, 2\pi)$. Then, using this in (D.3) and performing the integration, one recovers the representation of $\mathcal{P}_{Q(\alpha)} : D(M) \to D(M)$.

$$\langle \ell_1 m_1 | \mathcal{P}_{Q(\alpha)} | \ell_2 m_2 \rangle = \frac{\sin(\alpha m_1)}{\alpha m_1} \delta_{\ell_1,\ell_2} \delta_{m_1,m_2},$$
(D.4)

and we finally obtain the exact form of the transfer operator

$$\mathcal{F} = \mathcal{P}_{R_x(\gamma)} \mathcal{P}_{Q(\alpha)} \mathcal{P}_{R_x(\gamma)},$$
(D.5)

and this implies that the latter is a Frobenius-Perron operator of the local phase-space map $f : M \to M$:

$$f = R_x(\gamma) Q(\alpha) R_x(\gamma), \quad \mathcal{F} \equiv \mathcal{P}_f.$$
(D.6)

We have managed to obtain the exact form of the transfer operator, in both the density space and pointwise map in phase space, and as we can see in (D.2), it is block-diagonal in the total angular momentum $\ell$. The results for $\mathcal{F}_-$ can be found using the middle point reflection $\beta, \gamma \to \gamma, \beta$.

# E  Representation of $\mathcal{P}_{\Phi_{(\alpha,\beta,\gamma)}}$ in spherical harmonics

In this Appendix, we present the calculation of the matrix elements of the Frobenius-Perron operator $\mathcal{P}_{\Phi_{(\alpha,\beta,\gamma)}}$ of the local gate in the basis of spherical harmonics. We denote this basis as $|\ell, m\rangle \to Y_{\ell,m}$ $\ell = 0, \dots, \infty$, $|m| \le \ell$, which is clearly orthonormal with respect to the inner product (6):

$$\langle \ell_1 m_1 | \ell_2 m_2 \rangle = \int_{\mathcal{S}^2} d\vec{X} \, Y_{\ell_1 m_1}^*(\vec{X}) Y_{\ell_2 m_2}(\vec{X}) = \delta_{\ell_1,\ell_2} \delta_{m_1,m_2}.$$
(E.1)

As follows from (22), the local gate is composed of local single-site rotations $R_x(\theta)$, $\theta \in [0, 2\pi)$, together with the Ising Swap gate $I_\alpha$. The rotations around the $x$-axis are trivially known in the basis of spherical harmonics as $\mathcal{P}_{R_x(\theta)} = D(-\pi/2, \theta, \pi/2)$ where $D$ is the Wigner-D matrix [36] and is block diagonal in $\ell$ ($\langle \ell_1 m_1 | D | \ell_2 m_2 \rangle = 0$ when $\ell_1 \ne \ell_2$). Thus, we only require the representation of $I_\alpha$. Our approach is based on finding the kernel of the Ising gate on $\mathcal{S}^2 \times \mathcal{S}^2$, and then using this result to obtain its representation on $|\ell, m\rangle$.

We already know from (23), how $I_\alpha$ acts on two spins and this leads to the following kernel

$$\mathcal{P}_{I_\alpha}(\vec{X}_1, \vec{X}_2, \vec{X}_3, \vec{X}_4) = \delta(\vec{X}_1 - R_z(\alpha z_3)\vec{X}_4) \, \delta(\vec{X}_2 - R_z(\alpha z_4)\vec{X}_3),$$
(E.2)

(one should mention that we choose polar coordinates $\vec{X}_i = (z_i, \varphi_i)$ $i = 1, \ldots, 4$ for the parametrization of the unit sphere). This operation couples two spins and thus by using (6) in the basis of two-spherical harmonics we obtain

$$\langle \ell_1 m_1, \ell_2 m_2 | \mathcal{P}_{I_\alpha} | \ell_3 m_3, \ell_4 m_4 \rangle = \delta_{m_1, m_4} \delta_{m_2, m_3}$$
$$\times \int_{\mathcal{S}^2} d\vec{X}_3 d\vec{X}_4 \, Y^*_{\ell_1 m_1}(R_z(\alpha z_3)\vec{X}_4) Y_{\ell_4 m_4}(\vec{X}_4) \, Y^*_{\ell_2 m_2}(R_z(\alpha z_4)\vec{X}_3) Y_{\ell_3 m_3}(\vec{X}_3). \quad \text{(E.3)}$$

The Kronecker deltas come from the integration over $\varphi_3, \varphi_4$, and in the expression above we can see the coupling of the rotations with the $z$-component of each other's vector. In order to continue our calculation, we have to note that a rotation around the $z$-axis is a translation over the azimuthal angle, so that the spherical harmonics satisfy $Y_{\ell,m}(R_z(\theta)\vec{X}) = Y_{\ell,m}(\vec{x})e^{im\theta}$. Based on this property one can decouple the $z$-components in the following way

$$\langle \ell_1 m_1, \ell_2 m_2 | \mathcal{P}_{I_\alpha} | \ell_3 m_3, \ell_4 m_4 \rangle$$
$$= \int_{\mathcal{S}^2} d\vec{X}_3 d\vec{X}_4 \, Y^*_{\ell_1 m_1}(R_z(\alpha \frac{m_2}{m_1} z_4)\vec{X}_4) Y_{\ell_4 m_4}(\vec{X}_4) \, Y^*_{\ell_2 m_2}(R_z(\alpha \frac{m_1}{m_2} z_3)\vec{X}_3) Y_{\ell_3 m_3}(\vec{X}_3). \quad \text{(E.4)}$$

At this point, we have managed to couple the rotations $R_z$ of one spin with its own $z$-component. This type of nonlinear rotation is called 'torsion' $T(a)\vec{X} = R_z(az)\vec{X}$ (where $a$ is the coupling constant with $T(0) = \mathbb{1}$) and its representation in spherical harmonics has been obtained in [37]. In particular, it was found that

$$\langle \ell m | \mathcal{P}_{T(a)} | \ell' m' \rangle = \delta_{m,m'}(-1)^m \sqrt{(2\ell+1)(2\ell'+1)} \sum_{p=|\ell-\ell'|}^{\ell+\ell'} (-i)^p j_p(-ma) C^{\ell \ell' p}_{000} C^{\ell \ell' p}_{-mm0}, \quad \text{(E.5)}$$

where $j_p$ is the spherical Bessel function and $C^{\ell_1, \ell_2, \ell_3}_{m_1, m_2, m_3}$ are the Clebsch-Gordan coefficients. Finally, we obtain the representation of the Ising Swap gate

$$\langle \ell_1 m_1, \ell_2 m_2 | \mathcal{P}_{I_\alpha} | \ell_3 m_3, \ell_4 m_4 \rangle = \langle \ell_1 m_1 | \mathcal{P}_{T(\alpha \frac{m_2}{m_1})} | \ell_4 m_1 \rangle \, \langle \ell_2 m_2 | \mathcal{P}_{T(\alpha \frac{m_1}{m_2})} | \ell_3 m_3 \rangle \, \delta_{m_1, m_4} \delta_{m_2, m_3}.$$
$$\text{(E.6)}$$

This expression is valid also in the case when $m_1, m_2 = 0$ since, as we can observe from (E.5), the denominators cancel in the argument of $j_p$. Now, we only need to combine all of the above, which leads to the following representation

$$\langle \ell_1 m_1, \ell_2 m_2 | \mathcal{P}_{\Phi_{\alpha,\beta,\gamma}} | \ell_3 m_3, \ell_4 m_4 \rangle$$
$$= \sum_{q_1, q_2 = -\ell_1, -\ell_2}^{\ell_1, \ell_2} \langle \ell_1 m_1 | \mathcal{P}_{R_x(\beta)} | \ell_1 q_1 \rangle \langle \ell_4 q_1 | \mathcal{P}_{R_x(\beta)} | \ell_4 m_4 \rangle \langle \ell_2 m_2 | \mathcal{P}_{R_x(\gamma)} | \ell_2 q_2 \rangle \langle \ell_3 q_2 | \mathcal{P}_{R_x(\gamma)} | \ell_3 m_3 \rangle$$
$$\times \langle \ell_1 q_1 | \mathcal{P}_{T(\alpha \frac{q_2}{q_1})} | \ell_4 q_1 \rangle \, \langle \ell_2 q_2 | \mathcal{P}_{T(\alpha \frac{q_1}{q_2})} | \ell_3 q_2 \rangle. \quad \text{(E.7)}$$

# F The modes which contribute to the correlations

In this Appendix, we prove, that the only contributing $\ell$-subspaces to the correlations are the common ones of the expansions of the observables over the spherical harmonics. We denote these subspaces as $V^\ell = span(\{|\ell, m\rangle\}_{m=-\ell}^{\ell})$. The proof is a consequence of the block diagonal form of $\mathcal{F}$ ($\equiv \mathcal{F}_+$). Specifically, the transfer operator is block diagonal in $\ell$, meaning, that it is the direct sum $\mathcal{F} = \bigoplus_{\ell=0}^{\infty} \mathcal{F}^\ell$, where $\mathcal{F}^\ell$ are the blocks of each total angular momentum subspace.

It is thus convenient to work in the picture where the Hilbert space $L^2(\mathcal{S}^2) = \overset{\infty}{\underset{\ell=0}{\oplus}} V^\ell$ is a direct sum of the total angular momentum subspaces. Now according to this picture, the two local observables mentioned in the main text would also be decomposed as $|a\rangle = \overset{\infty}{\underset{\ell=0}{\oplus}} |a^\ell\rangle$, $|b\rangle = \overset{\infty}{\underset{\ell=0}{\oplus}} |b^\ell\rangle$. Assume, that their expansions over the spherical harmonics overlap only with a finite number of $V^\ell$ spaces, which we denote as $\ell_i^a \quad i = 1, \ldots, n_a$ and $\ell_j^b, \quad j = 1, \ldots, n_b$ respectively. The integers $n_a, n_b$ are the total number of overlapping $V^\ell$ of the observables. This would imply that the components $|a^\ell\rangle, |b^\ell\rangle$ vanish trivially at the rest of the total angular momentum subspaces:

$$
\begin{aligned}
|a^\ell\rangle &= \vec{0}_\ell, \quad \text{for} \quad \ell \neq \ell_i^a, \\
|b^\ell\rangle &= \vec{0}_\ell, \quad \text{for} \quad \ell \neq \ell_j^b,
\end{aligned}
\tag{F.1}
$$

where $\vec{0}_\ell$ is the zero vector in $V^\ell$. Moreover, in this picture, the Hermitian product splits into a sum of Hermitian products over $V^\ell$ and by using $|\circ\rangle = |1\rangle/2\sqrt{\pi}$, we obtain from (19)

$$
C_{a,b}(t,t) = \frac{1}{4\pi}\Big( \sum_{\ell=0}^\infty \langle a^\ell|(\mathcal{F}^\ell)^{2t}|b^\ell\rangle - \frac{1}{4\pi}\langle 1|b\rangle\langle 1|a\rangle \Big) = \frac{1}{4\pi}\sum_{\ell_c \neq 0} \langle a^{\ell_c}|(\mathcal{F}^{\ell_c})^{2t}|b^{\ell_c}\rangle, \tag{F.2}
$$

where we applied (F.1) and now, one can observe, that the only non-vanishing terms are the ones of the common subspaces $\ell_c$ between $\ell_i^a$ and $\ell_j^b$. The space $V^0$ of the constant on $\mathcal{S}^2$ scalars, does not contribute to the correlations since it is being cancelled out from the second term in (F.2). In addition, our result automatically implies, that only the eigenvalues of $\mathcal{F}^{\ell_c}$ contribute and thus the exact 2-point function is defined by a finite set of exponentials. One can obtain the results for the other chirality of correlations by using the middle point reflection $\beta, \gamma \to \gamma, \beta$.

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
