# Peer review of "Dual symplectic classical circuits: An exactly solvable model of many-body chaos"

_SciPost Physics, doi:SciPost Phys. 16, 049 (2024)_

## Round 1 · Referee Report · Anonymous (Referee 1) · 2023-8-28

Strengths

  1. Generalizes ideas from dual unitary circuits to classical deterministic ones
  2. Proves several exact results, in particular for two-point correlators
  3. Brings methods more commonly used in quantum dynamics (circuits, tensor networks) to classical many body
  4. Shows explicitly (and in full detail for a class of examples) how deterministic overall dynamics contracts to stochastic dynamics for a subset of degrees of freedom (2-pt correlators in this case)

Weaknesses

None

Report

This is an important paper which adds to the growing corpus of work on "dual" circuit dynamics, an area of much current study started by one of the authors and coworkers. While most work has been on quantum circuits with dual unitary dynamics (i.e. unitary both for time and space propagation) these ideas can also be applied to classical systems. This is what this paper does by considering dual *symplectic* (i.e. phase space volume preserving) classical dynamics. The key property is that of (let's call it) unitality in both space and time, see eq.15, whereby time and space propagation leave the flat probability vector invariant (both forward and back). As for DU circuits, several results follow immediately. The focus of interest here is on two-point correlators: they are non-trivial only in rays, and their dynamics reduces to a stochastic one. These general and exact results are fleshed out for class of problems corresponding to coupled dynamics of spherical spins, for which many results are made explicit. One can immediately think of follow up questions to this work, so it is clearly opening up avenues for further research. Except for very minor changes (see below) the paper can be published as is.

Requested changes

  1. Montecarlo -> Monte Carlo in the abstract.

  2. Please clarify what the "hooks" are meant to be at the right and left edges of Fig.2.

  3. Present the Appendices in the same order as they are mentioned in the main text.

  • validity: high
  • significance: high
  • originality: high
  • clarity: high
  • formatting: excellent
  • grammar: excellent

Author:  Alexios Christopoulos  on 2023-11-27  [id 4147]

(in reply to Report 1 on 2023-08-28)

See the attached document for the changes to the manuscript and answers to the reviewers.

Attachment:

resubmission_letter.pdf

Author:  Alexios Christopoulos  on 2023-09-06  [id 3956]

(in reply to Report 1 on 2023-08-28)

Thank you very much for your comments and requested changes. I am going to take them into consideration. The "hooks" or else the curly edges of the graph in Fig.2 indicate the periodic boundary conditions in space for our system. Following your suggestion, I am going to clarify this in the caption of the respective figure.

---

## Round 1 · Referee Report · Anonymous (Referee 2) · 2023-9-13

Report

In this work, the authors study a special type of classical dynamical system known as dual-symplectic circuit. It consists of a discrete set of classical variables updated using local rules in a discrete fashion, similarly to classical cellular automata. In addition to symplecticity, these dynamics are also symplectic "in the rotated channel". This is a very strong mathematical property that allows for the derivation of analytic results. This fact is remarkable because the model is not integrable in the traditional sense.

This set of models, which are natural classical versions of so-called dual-unitary circuits previously studied, were not introduced in this paper. However, the authors develop a general formalism to compute analytically dynamical correlation functions, similar to the quantum case. The general formalism is nicely applied to one specific model (the Ising swap model) and the predictions are tested quantitatively against numerical Montecarlo data.

Overall, I think this is a strong paper. Although perhaps not extremely innovative, it fills a gap in the literature, by extending to the classical case the formalism previously developed in the context of dual-unitary quantum circuits. The draft is well written and all the predictions are supported by convincing numerics. I don't have any particular comment on how to improve the readability of the draft, so I recommend publication of the manuscript as is.

I have, however, just two questions/curiosities, and two comments about references.

First, I believe that, compared to the quantum case, this model could be useful to study more easily the periodicity of the classical orbits and how this depends on integrability/non-integrability of the model. It would be interesting to study in particular the scaling of the orbit sizes with the system size. Do the authors have some intuition about this aspect?

Second, would it be possible to also generalize, say, tri-unitary quantum circuits (as introduced in https://journals.aps.org/prresearch/abstract/10.1103/PhysRevResearch.3.043046) to the classical case?

Regarding the literature, when stating

"Interestingly, dual unitary quantum circuits can exhibit strongly chaotic quantum dynamics, whose classical simulation is in general expected to be exponentially hard in system size"

I believe it would be important to cite

https://quantum-journal.org/papers/q-2022-01-24-631/

where a rigorous result along these lines was proven.

Finally, two very trivial comments. It appears T. Prosen is misspelled (I believe?) as "T. c. v. Prosen" in multiple entries in the literature. Also, journal names appear sometimes in short-hand notation (e.g. Phys. Rev. Lett.), sometimes with full names (e.g. Physics Letters A). The authors might want to make the notation uniform.
  • validity: -
  • significance: -
  • originality: -
  • clarity: -
  • formatting: -
  • grammar: -

Author:  Alexios Christopoulos  on 2023-11-27  [id 4149]

(in reply to Report 2 on 2023-09-13)

See the attached document for the changes to the manuscript and answers to the reviewers.

Attachment:

resubmission_letter_Eg0bL7H.pdf

---

## Round 1 · Referee Report · Pieter W. Claeys (Referee 3) · 2023-9-15

Strengths

1- Exactly solvable models of classical dynamics 2- Natural generalization of dual-unitarity to classical dynamics 3- Detailed analysis and exact reuslts for a representative example

Weaknesses

1- No explicit discussion of notions of chaos/integrability

Report

In this work, the authors introduce the notion of "dual symplectic classical circuits" as the classical analogue of "dual unitary quantum circuits", which were recently introduced as exactly solvable models of quantum chaos in which certain dynamical properties (e.g. correlation functions) can be calculated exactly. These calculations are often done graphically, and the authors here show how demanding symplecticity in the time and space direction results in identical graphical identities, from which the calculations from dual-unitarity can be extended to classical circuits.

Specifically, it is shown that in these models two-point correlation functions vanish everywhere except on the edge of the causal light cone, where they can be calculated using a transfer operator formalism. In the specific case of an Ising Swap model with additional single spin rotations, the authors explicitly analyse this transfer operator. It is shown that the transfer operator conserves total angular momentum and various exact results on the eigenspectrum are presented, including autocorrelation functions for the $S_z$ spin component.

The results are interesting, convincing, and well presented. Furthermore, there are various results in the growing literature on dual-unitarity that can subsequently be studied in these classical circuits, such that this work opens up new pathways in this research direction. As such, I am happy to recommend this work for publication in SciPost Physics provided some questions/comments are addressed.

Requested changes

1- One of the most remarkable properties of dual-unitary circuits is that it could be explicitly shown that these circuits satisfy the usual 'definition' of quantum chaos, with major contributions from one of the authors. However, even though the title of this work is "An exactly solvable model of many-body chaos", there is no discussion about whether or not these circuits are chaotic (although the ergodicity is discussed). Since classical chaos is well defined, it would be interesting if the authors could explicitly comment on the chaotic properties of their analysed Ising swap model. Lyapunov exponents are mentioned in the introduction, but then not discussed in the main text. In a related comment, classical and quantum notions of integrability are defined differently, and it would be interesting if the authors could comment on how these notions are satisfied in the presented model when mentioning the integrable points of the Ising swap gate.

Some minor comments 2- On page 3, it is mentioned that the spectrum of the Jacobian needs to include pairs of eigenvalues and that the Lyapunov exponents appear in pairs. Could the authors provide either an argument or a reference as to why that is the case? 3- In the same section, when introducing the model, the authors write "symplectic maps always involve $d$-pairs of conjugate variables,the configuration $q$ and the momentum $p$, which can be seen as the coordinates of a $2d$ dimensional manifold", before discussing such properties as those in my previous point. However, in the specific example considered in this manuscript the authors consider the dynamics of spin variables. It might be worthwhile to also discuss/mention spin variables in that section and the connection with typical pairs of conjugate variables. 4- Small typo below Eq. (25): "The transfer operator is just the Perron–Frobenius of ..." 5- Conservation of total angular momentum typically follows from rotational symmetry, and it might be useful to discuss this symmetry in the main text. Right now it is mentioned below Eq. (B.6) that "This is not surprising, since as we can see from (B.3),(B.6) the transfer operator is just a composition of rotations, which preserve the total angular momentum." It would be useful to include such a discussion in the main text.

  • validity: top
  • significance: high
  • originality: good
  • clarity: high
  • formatting: good
  • grammar: good

Author:  Alexios Christopoulos  on 2023-11-27  [id 4150]

(in reply to Report 3 by Pieter W. Claeys on 2023-09-15)

See the attached document for the changes to the manuscript and answers to the reviewers.

Attachment:

resubmission_letter_Ug6bl8v.pdf

---

## Round 2 · Referee Report · Anonymous (Referee 2) · 2023-11-27

Report

The authors have addressed my comments fully, and I thus recommend publication of the manuscript.

---

## Round 2 · Referee Report · Pieter W. Claeys (Referee 3) · 2023-11-28

Report

All my questions and comments have been appropriately addressed and I am happy to recommend this paper for publication in SciPost Physics.

---

## Round 2 · Referee Report · Anonymous (Referee 1) · 2023-11-29

Report

I am happy with the revised version which in my view can be published as is.

---

## Round 2 · Author Response

Dear editor,

Herewith we resubmit our manuscript titled “Dual symplectic classical circuits: An exactly solvable model of many-body chaos” to SciPost. We would like to express our gratitude to the reviewers for their time and their invaluable feedback on the initial submission. In response to their constructive comments, we have made improvements to our work, enhancing its scientific rigour and
clarity. Please see our response below.

Yours sincerely,

Alexios Christopoulos (on behalf of all coauthors)

For more details regarding the changes made to the manuscript and the replies to the reviewers, please refer to the attached PDF as a response to the reviewers' comments.

---

## Round 2 · List of Changes

For more details regarding the changes made to the manuscript and the replies to the reviewers, please refer to the attached PDF as a response to the reviewers' comments.

---

## Editorial Decision

published